# Protein Recovery of Tra Catfish (*Pangasius hypophthalmus*) Protein-Rich Side Streams by the pH-Shift Method

**DOI:** 10.3390/foods11111531

**Published:** 2022-05-24

**Authors:** Hang Thi Nguyen, Huynh Nguyen Duy Bao, Huong Thi Thu Dang, Tumi Tómasson, Sigurjón Arason, María Gudjónsdóttir

**Affiliations:** 1Faculty of Food Science and Nutrition, University of Iceland, Aragata 14, 102 Reykjavik, Iceland; sigurjar@hi.is (S.A.); mariagu@hi.is (M.G.); 2Faculty of Food Technology, Nha Trang University, 02 Nguyen Dinh Chieu, Nha Trang 650000, Vietnam; hndbao@ntu.edu.vn (H.N.D.B.); dangthithuhuong@ntu.edu.vn (H.T.T.D.); 3UNESCO GRÓ—Fisheries Training Programme, Fornubudum 5, 220 Hafnarfjordur, Iceland; tumi@groftp.is; 4Matis, Icelandic Food and Biotech R&D, Vinlandsleid 12, 113 Reykjavik, Iceland

**Keywords:** Tra catfish, dark muscle, cut-offs, protein isolate, protein, lipid removal, amino acids

## Abstract

Increasing protein demand has led to growing attention being given to the full utilization of proteins from side streams in industrial fish processing. In this study, proteins were recovered from three protein-rich side streams during Tra catfish (*Pangasius hypophthalamus*) processing (dark muscle; head-backbone; and abdominal cut-offs) by an optimized pH-shift process. Physicochemical characteristics of the resulting fish protein isolates (FPIs) were compared to industrial surimi from the same raw material batch. The pH had a significant influence on protein extraction, while extraction time and the ratio of the extraction solution to raw material had little effect on the protein and dry matter recoveries. Optimal protein extraction conditions were obtained at pH 12, a solvent to raw material ratio of 8, and an extraction duration of 150 min. The resulting FPI contained <10% of the fat and <15% of the ash of the raw material, while the FPI protein recovery was 83.0–88.9%, including a good amino acid profile. All FPIs had significantly higher protein content and lower lipid content than the surimi, indicating the high efficiency of using the pH-shift method to recover proteins from industrial Tra catfish side streams. The FPI made from abdominal cut-offs had high whiteness, increasing its potential for the development of a high-value product.

## 1. Introduction

Fish is a limited resource, and the global demand for fish protein is increasing faster than can be met with traditional resources [1,2,3]. Therefore, recovering protein for human food from side streams in industrial fish processing is of great interest. Side streams can be used directly as food or processed into fish-based protein foods such as sausages, cakes, snacks, sauces, or other products such as gelatin, dietetic products, pharmaceuticals, natural pigments, cosmetics, or constituents in other products [4,5,6]. The improved utilization of fish side streams may thus increase products for human consumption and add value.

Proteins are found in all parts of fish, but different side streams may have different protein content and composition [7,8]. There are three main categories of proteins in fish: structural proteins, sarcoplasmic proteins, and connective tissue proteins, which have different physiochemical properties [9]. About 30–50% of the muscle is usually left on the frame during filleting [9]. These muscle proteins are highly nutritious and digestible, and flesh from frames and abdominal-cut offs can be used to produce mince, surimi, or surimi-based products [10]. There are two types of muscle in finfish: white and dark muscle. The dark muscle is used for continuous swimming action, while the white muscle supplies rapid energy bursts. Due to high levels of pro-oxidants, such as heme protein and iron, and its often-unwanted colour, the dark muscle is commonly removed from the fillets in the trimming step of fatty fish processing [11,12,13,14]. However, the dark muscle is used to produce fishmeal, pet food, fish silage, and fertilizer at a low economic value. Nevertheless, as it contains high-quality proteins and bioactive protein-derived compounds, it also has potential for use in food and healthcare products [15,16,17]. Several studies have been carried out on processing the dark muscle from several species into functional proteins such as hydrolysates, isolates, and peptides [16,17,18].

Tra catfish are commonly farmed in Southeast Asia, including Vietnam, Thailand, Indonesia, and Cambodia. They are currently the most important farmed freshwater species in Vietnam [19,20]. Globally, Vietnam is the third-largest exporter of fish and fish products, with most of its revenues coming from farmed Tra catfish. Vietnam is also the biggest producer of Tra catfish, with nearly 1.42 million tonnes in 2018 [21]. Over 97% is processed into frozen, white fillets for export [22]. However, only about 40% of the fish processed is used for human consumption [23]. The growth in Tra catfish production means an increase in its side streams (head, backbone, skin, cut-offs, trimmings, and viscera), which account for 62–67% of production [24,25]. The side streams from the production of fillets thus amounted to over 800,000 tonnes in 2018 [19]. Traditionally, these side streams have mostly been used to produce fishmeal and silage which are of low economic value and generate limited profits [24,25,26,27,28]. Some of the fishmeal produced is used to produce feed for Tra catfish which does not require a high proportion of animal protein in its diet [29]. Recently, the industry is producing more fishmeal than needed for feed production. The large use of fishmeal in aqua-feeds has been reported as a significant environmental impactor, causing global warming and eutrophication [30]. Fishmeal and fish oil in their diets can be significantly replaced by various alternative feed ingredients from plant origin, insects, microalgae, microbial proteins, and seaweed, which have lower prices. Another common use of Tra catfish fishmeal is for tilapia, shrimp, poultry and pig feeds [29]. Some of these side streams have, in recent years, been utilized for producing surimi, although the surimi industry started in Vietnam in the 1990s [31]. However, the processing industry has the potential to make more profit from the production of protein isolates and hydrolysates used in food ingredients in value-added products, and/or for feed intended for juvenile fish and other farmed animals that require protein-rich and highly digestible diets. The utilization of these side streams could thus bring profit to processing companies far beyond the margins of selling fish fillets and only recycling the side streams within the industry, as is the current situation.

Side streams have high lipid and ash content, ranging from 15.3 to 29.8% and 2.4 to 5.7%, respectively [19], which can limit their direct use as human foods. Several authors have studied the use of Tra catfish side streams to recover proteins [19,32,33,34,35,36]. Still, there has been limited work completed on the use of dark muscle and on comparing the properties of FPIs recovered from different side streams. In order to maximize the utilization of individual side streams, the physicochemical characteristics of different side streams should be studied.

The pH-shift method is commonly used to recover or isolate proteins from fish side streams [17,37]. The preparation of fish protein isolates includes three main steps: (i) the solubilization of proteins at low or high pH (≤3.5 or ≥10.5); (ii) the removal of fat and other impurities by using a high-speed centrifuge; and (iii) the precipitation of the protein at their isoelectric point (pH = 5.5) [37]. FPI can be frozen for later use, such as for surimi or mince production [38]. However, as mentioned earlier, different side streams may have specific water, protein, lipid, ash, and pigment composition. Fish proteins from side streams may also be contaminated by other tissues, such as the skin, backbone, or blood [37], which can be barriers to successful protein recovery from the side streams and may challenge the stability of the protein product during storage. High lipid and pigment content, such as haemoglobin and myoglobin from the blood, may cause rancid, fishy odours in the final product [37]. Blood contamination often occurs in current production systems where side streams are commonly processed together. However, processing FPIs from each side stream separately could potentially increase the quality and value of the final products.

In this study, FPI was produced from the dark muscle, head and backbone blends, and abdominal cut-offs using the pH-shift method. The proximate composition, colour and protein composition, and properties of the different FPIs and surimi produced from the same processing side stream batches were analysed and compared.

## 2. Materials and Methods

### 2.1. Raw Materials and Sampling

#### 2.1.1. Collection of Side Streams

A flowchart of the industrial Tra catfish filleting process is presented in Figure 1. The fish were bled and the pelvic fins manually removed before filleting. The rest raw materials from the filleting were separated into ***heads*** (also consisting of some flesh)***, backbones*** (including the tail, fins, and some skin and muscle), ***viscera,*** and ***pelvic fins***. The untrimmed fillets were then passed through a **skin** removal system (Nam Dung, Ho Chi Minh, Vietnam). The skinned fillets were then transferred to the trimming area, which had a room temperature of 18 ± 2 °C. Fat edges (mainly subcutaneous fat and some flesh) removed from the skinned fillets were considered ***trimmings***. Thereafter, the ***dark muscle*** of the fillet was removed manually, and the belly flap was cut off the trimmed and skinned fillets to obtain the ***final fillets*** and ***abdominal cut-offs.***

Dark muscle, head and backbone blend (HBB), and abdominal cut-offs (ACO) were collected and measured separately for this study (see the further list of abbreviations in Appendix A). Each sample was minced evenly using a meat grinder (TA57D, Didacta, Torino, Italia) in a cool room (16 ± 2 °C). The minced samples were then block frozen (5 kg per block) in a contact freezer (Mycom, Tokyo 135-8482, Japan) for 3 h at −35 to −40 °C, reaching a core temperature of −18 °C to −20 °C. The frozen blocks were then cut into 0.5–1 kg pieces appropriate for chemical analyses and trials. The pieces were individually packaged in polyethylene bags, and then put in styrofoam boxes for transport to the laboratory by a cold truck for 8 h. Upon arrival, the samples were stored at −25 ± 1 °C until they were analysed and were used within three months. Prior to use, samples were left in a refrigerator at 2–4 °C for 24 h to thaw.

#### 2.1.2. Chemicals

All chemicals used in the study were of analytical grade and purchased from Sigma-Aldrich (Missouri, TX, USA) and Merck (Darmstadt, Germany).

#### 2.1.3. Preparation of Protein Isolate from the Dark Muscle

The production of FPI using the pH-shift method [39] was performed at different ***pH*** values, ***extraction ratios,*** and ***extraction times*** of the protein solubilization step in order to find the optimal FPI production settings. A flow chart for the optimization of protein recovery from dark muscle is shown in Figure 2.

First, the pH of the protein solubilization was optimized during Trial 1 (Figure 2). Different amounts of either 2N HCl or 2N NaOH solutions were added to distilled water to obtain ***protein extract solutions (PES)*** with acidic pH 2, 3, 4 and alkaline pH 9, 10, 11, 12, and 13, adjusting the pH using a pH meter (Portavo 904 X, Knick, Berlin, Germany). About 200 g of minced dark muscle was homogenized (ULTRA- TURAX^®^ T18 basic, IKA, Germany) with 1600 mL of cold PES for 1 min, and the pH of the mixture was adjusted with 2N HCl and 2N NaOH solutions to obtain the targeted pH. The blends were kept in a fridge for 60 min at 0–4 °C for protein solubilization. The cold suspension was then centrifuged at 3000 rpm for 20 min at 4 °C (MF 600, Biobiz, Incheon, Korea). The supernatant (***protein solution***) was weighed and determined for protein content using the Bradford method [40] to evaluate the ***protein extractable recovery (PER)***. PER was also measured at pH 5. The protein solution was then adjusted to pH 5.5 (isoelectric protein point (PI)) using 2N NaOH and 2N HCl to precipitate the protein [41,42]. The aggregated precipitates were filtered through a nylon monofilament bag (mesh size: 25 micron; Dong Son Ltd., Vietnam), and the mixture was washed with distilled deionized water to a neutral pH. The mixture was then centrifuged at 3000 rpm for 20 min to dewater and remove the NaCl to form the final ***fish protein isolate (FPI)***. The FPI was weighed and measured for water and protein contents to evaluate the ***protein recovery (FPI-PR)*** and ***dry matter recovery*** (FPI-DMR), as described in the analysis section.

Trial 1 indicated that pH 12 was optimal for the solubilization step of the FPI production and was, therefore, selected for the subsequent trials. The effects of using different ratios between PES and dark muscle (***extraction ratio***, *v*:*w*) were evaluated, with ratios ranging between 5 and 10 times (*v*:*w*). The PER, FPI-PR, and FPI-DMR were determined as above.

During Trial 2, a ratio between PES and the raw material of 8 (*v*:*w*) was found to be optimal for protein extraction and was thus used in further experiments. The FPI was then produced at ***different extraction times***, including 30, 60, 90, 120, 150, and 180 min in Trial 3. The PER, FPI-PR and FPI-DMR were investigated in the same procedures as above. Each experiment was performed in triplicate.

#### 2.1.4. FPI Preparation from Different Side Streams at the Optimal Conditions

The protein isolates were produced from the HBB and ACO at the above-optimized conditions. FPI-PR, FPI-DMR, lipid and ash removal effectivity were compared at the previously determined optimal conditions for all protein isolates made from dark muscle, HBB, and ACO. In addition, quality properties, including proximate composition, amino acid profiles, protein patterns, and colour of the protein isolates, were evaluated and compared with commercial surimi produced at the company from the same batch of rest materials.

### 2.2. Analyses

#### 2.2.1. Proximate Composition

Water content of the samples was determined according to ISO 6496:1999. Approximately 5.0 g of sample was placed in a small porcelain bowl. The bowl was dried in an oven at 103 ± 1 °C for 4 h, then allowed to cool to ambient temperature for about 30 min in a desiccator before the weight was recorded again.

The crude protein content of the samples was measured using the Kjeldahl method according to ISO 5983-2:2008. About 2 g of minced sample was digested in 17.5 mL concentrated sulphuric acid with copper sulphate added as a catalyst at approximately 420 °C. The digested mixture was made alkaline with NaOH, and the nitrogen was distilled off as NH_3_. The NH_3_ was “trapped” in a 1% boric acid solution. The amount of ammonia nitrogen in the solution was measured by titration with a standardized H_2_SO_4_ solution. A nitrogen conversion factor of 6.25 was used to calculate crude protein content.

Lipids were measured according to the Bligh and Dyer [43] method. About 25 g of the sample was homogenized with 50 mL of chloroform, 50 mL of methanol and 25 mL of 0.88% KCl for 4 min. The mixture was centrifuged at 4 °C for 20 min at 2500 rpm. The chloroform fraction (the liquid bottom part) which contained the lipids was then collected and filtrated through a glass microfiber filter under vacuum suction. Exactly 2 mL of the chloroform fraction was pipetted into a glass tube and placed in a vacuum dryer at 55 °C to remove the chloroform solvent. The remaining mixture was weighed to measure the total lipid content.

The ash content was determined according to the method of the Association of Official Analytical Chemists (AOAC, 2000). About 5 g of the sample was weighed into a crucible. The sample was heated overnight at 550 ± 3 °C and then cooled down in a desiccator before being weighed. Ashes were quantified gravimetrically. The water, crude protein, lipid and ash contents were expressed as a percentage of wet weight.

#### 2.2.2. Protein Extractable Recovery (PER), FPI Protein, and FPI Dry Matter Recoveries

Protein extractability was defined as a percentage of the protein extracted into the solution during the solubilization step compared to the initial raw material protein, calculated using Equation (1).
(1)$$\mathrm{PER}\;(\%)=\frac{\mathrm{Protein}\;\mathrm{content}\;\mathrm{of}\;\mathrm{the}\;\mathrm{protein}\;\mathrm{solution}}{\mathrm{Protein}\;\mathrm{content}\;\mathrm{of}\;\mathrm{the}\;\mathrm{raw}\;\mathrm{material}}\;\times \;100$$

FPI protein recovery was defined as the recovered protein amount compared to the protein content of the initial raw material, as calculated by Equation (2).
(2)$$\mathrm{FPI}\text{-}\mathrm{PR}\;(\%)=\frac{\mathrm{Protein}\;\mathrm{content}\;\mathrm{of}\;\mathrm{the}\;\mathrm{FPI}}{\mathrm{Protein}\;\mathrm{content}\;\mathrm{of}\;\mathrm{the}\;\mathrm{raw}\;\mathrm{material}}\;\times \;100$$

The protein content of the solution was measured using the Bradford method [40]. Exactly 50 μL of soluble protein sample was mixed with 2.5 mL of Bradford reactive solution, then incubated for 25 min at ambient temperature. The absorbance was read at 595 nm in a DR6000 UV-VIS spectrophotometer (HACH, Düsseldorf, Germany). The protein content was calculated using a standard curve made with bovine serum albumin with concentrations ranging between 0.1 and 1.4 mg/mL.

The protein content of the FPIs and the initial raw material was extracted according to Mæhre et al. [44]. Approximately 1 g sample was homogenized with 60 mL of 0.1N NaOH in 3.5% NaCl solution. The mixture was then incubated for 90 min at 60 °C following centrifugation at 5000 rpm at 4 °C for 30 min (TJ-25 Centrifuge, Beckman Coulter, CA, USA). The supernatants were measured for protein content using the Bradford method, as described above.

FPI-DMR was defined as the dry matter content recovered compared to the dry matter from the initial raw material during the FPI production, as calculated according to Equation (3).
(3)$$\mathrm{FPI}\text{-}\mathrm{DMR}\;(\%)=\frac{\mathrm{Dry}\;\mathrm{matter}\;\mathrm{content}\;\mathrm{of}\;\mathrm{the}\;\mathrm{FPI}}{\mathrm{Dry}\;\mathrm{matter}\;\mathrm{content}\;\mathrm{of}\;\mathrm{the}\;\mathrm{raw}\;\mathrm{material}}\;\times \;100$$

#### 2.2.3. Lipid and Ash Removal during FPI Production

Lipid and ash removal (%) during FPI production was assessed by the proportional weight loss of lipid/ash during the production compared to the lipid/ash amount in the raw material. The weight loss of lipids and ash during FPI production was calculated by the initial amount in the raw material minus the amount in the FPI produced.

#### 2.2.4. Amino Acid Analysis

The amino acid composition of the FPI samples was measured using the liquid chromatography–mass spectrometry (LC-MS) method according to ISO 13903:2005. Amino acids were liberated from the protein using an EZ: faast LC-MS kit system (Phenomenex, Torrance, CA, USA). An ion-exchange chromatography separated the individual amino acids through the EZ: faast AAA-MS column (250 mm × 2.0 mm, 4 µm) (Phenomenex, Torrance, CA, USA). They were then detected by ninhydrin reaction at λ 570 nm (λ 440 nm for proline) in the Shimadzu LC-MS 8030 system (Kyoto, Japan). Amino acid content was expressed as g/100 g crude protein.

#### 2.2.5. SDS-PAGE Pattern

Sodium dodecyl sulfate (SDS) slab gel electrophoresis with dimensions of 140 × 140 × 1 mm was performed according to the modified method of Laemmli [45] using a precast gel Mini-protean TGX 10% (Bio-Rad Lab., Inc., Hercules, CA, USA). The running buffer contained 3 g/L Tris base, 14.4 g/L glycine, and 1g/L SDS in deionized water (pH 8.3). About 5 g of the sample was extracted with 25 mL of 1M NaCl buffer (pH 7.0) and diluted to a 2 mg/mL concentration. Then, 10 μL of the protein extracts were mixed well with 10 μL 4x Laemmli SDS loading buffer (1610747, Bio-Rad Lab., Inc., CA, USA) containing β-mercaptoethanol and placed into a 96-well plate. The sample wells were then entirely covered by an appropriate tape piece and heated at 100 °C for 5 min in a PCR heating block (Bio-Rad Lab., Inc., CA, USA). Then, 10 μL of the mixture was loaded into the electrophoresis gel well. Electrophoresis was conducted in a Mini-protean^®^Tetra Cell (Bio-Rad Lab., Inc., CA, USA) coupled with an electrophoresis power supply (Pharmacia Biotech, Uppsala, Sweden) at a constant current of 30 mA per gel until the tracking blue dye front reached the end of the gel. The gels were collected and stained in a dye solution (containing 0.05% Coomassie blue, 25% isopropanol and 10% acetic acid) overnight. The gels were then de-stained with a 10% acetic acid solution until the gel background was clear for photography by a Canon photo scanner (Tokyo, Japan). The molecular weight of protein bands was estimated using the Spectra^TM^ multicolour broad range protein ladder (10-260 kDa) (Thermo Fisher Scientific, Waltham, MA, USA).

#### 2.2.6. Colour

The intensity of the colour was determined with a Minolta Chroma Mette CR-400 (Minolta, Osaka, Japan) using the CIE Lab system as described by Abdollahi et al. [46]. The instrument recorded the *L ** value (brightness) on a scale of 0 to 100 from black to white; the *a ** value (redness) from −60 to 60, where a > 0 represents the red component and a < 0 represents the green component; and the *b ** value (yellowness) from −60 to 60, where (+) stands for the yellow component and (−) stands for the blue component. The whiteness was calculated using Equation (4), as described by Surasani et al. [47]:(4)$$\mathrm{Whiteness}=100-\sqrt{{\left(100\;-\;L*\right)}^{2}+\;{a*}^{2}+{b*}^{2}}\;$$

#### 2.2.7. Statistical Analysis

All data summaries and statistical analyses were carried out in Microsoft Excel 365 (Microsoft Inc., Redmond, Washington, DC, USA) and IBM SPSS Statistics software (Version 22, IBM, 1 New Orchard Road, Armonk, New York, NY, USA). One-way analysis of variance (ANOVA), Tukey’s HSD tests, and Student’s t-tests were performed on means of the variables. All statistical analyses were performed assuming a significant difference set to the 5% level (*p* < 0.05).

## 3. Results and Discussion

### 3.1. Optimization of Protein Recovery from Dark Muscle

#### 3.1.1. Effects of pH on Extraction Levels (Trial 1)

The minimum PER was observed at pH 5 (17.9 ± 3.7%) and gradually increased as the pH moved up or down (Figure 3). Protein extraction recovery (PER) is affected by the protein solubility at a given pH, and the protein solubility depends on the electrostatic and hydrophobic interactions between the protein molecules. When electrostatic repulsion is higher than the hydrophobic interactions, protein solubility increases and vice versa [48]. Proteins become positively or negatively charged at a pH lower or higher than their PI. This leads to electrostatic repulsion between molecules and the hydration of charged residues (protein–water interactions increase). The PI of fish protein is commonly around 5.5 [37,49].

Protein extraction yield is greater under alkaline than acid conditions, as reported by Zayas (2012) and Abdollahi and Undeland [50]. This may be due to the denaturation of proteins during harsh acid treatment. Denatured proteins will be lost in the sediment after the centrifugation of the extraction mixture [51,52]. The highest PER was obtained at pH 12 (95.3 ± 3.3%) (*p* < 0.05). Higher PER also resulted in higher FPI protein recovery (FPI-PR) and FPI dry matter recovery (FPI-DMR), as expected. A similar effect of the extraction pHs on the PER, FPI-PR, and FPI-DMR was indicated in this study. The highest FPI-PR and FPI-DMR were obtained at pH 12, with values of 70.9 ± 4.8% and 30.3 ± 0.4%, respectively (*p* < 0.05). Although the protein recovery values at pH 11 were lower than the corresponding values at pH 12 (*p* < 0.05), the FPI-DMR was not significantly different between these two pH values (*p* > 0.05). These results reflect that the lipid that remained in the FPI at pH 11 was higher than at pH 12. This is in agreement with the study of Chen and Jaczynski (2007), which stated that the pH of extraction buffers can influence the lipid content of the final FPI. Alkaline washing has, furthermore, been found to be more effective in removing lipids than using water with a lower pH for washing [42].

The optimal pH for FPI production from the dark muscle occurred at pH 12, which is within the range obtained in a previous study, which stated an optimal pH of 11.5 and 12.5 for salmon and herring rest materials, respectively [50]. Several previous studies have reported the maximum solubility of fish muscle protein at pH 12, such as in tilapia frame rest materials [53] and salmon rest materials [50]. Nevertheless, this pH value was slightly different from the pH of 10~11 recommended by Kristinsson, Lanier, Halldorsdottir, Geirsdottir and Park [17], and the pH 13 as recommended by Rohu for rest materials [47].

#### 3.1.2. Effects of Using Different Proportions of Extract Solution (Trial 2)

Solubilization is performed using extraction solution in a 5–10 times ratio compared to the raw material [17]. PER increased slightly as the extraction solution and the raw material ratio increased from 5 to 6, and decreased again at a ratio of 9 (Figure 4). However, these differences in PER with different extraction ratios were not significant (*p* > 0.05). This could be due to low protein content in the raw material, so that even the lowest volume of the extract solution was enough to extract most of the proteins. However, the FPI-PR increased significantly at an extraction ratio of 8 (*p* < 0.05) compared to the lower ratios. However, no significant differences were seen in FPI-PR for extraction ratios between 5 and 7. This result is in agreement with Zhang and Chang [54], who showed that changing the ratio from 4 to 7 did not affect the FPI-PR produced from catfish by-products. The highest recovery rates were obtained at a ratio of 8 to 9 (70.9–73.2%).

The FPT-DMR was higher when the ratio of the extract solution was 8–10 than when it was 5–7 (*p* < 0.05), but was not significantly different within these ranges. Batista [55] suggested that a low ratio of extract solution may result in increased viscosity, limiting the effectiveness of the centrifugation and separation of solid particles, and resulting in more proteins being lost during the separation steps. In contrast, a high ratio can give a very diluted protein extract, reducing the recovery of protein in the precipitation and resulting in a decrease in FPI-PR at an extraction ratio of 10. The volume of water used for the fish protein isolate production plays an important role in industrial processing due to increased water use and discharge. Therefore, the ratio of 8 was chosen as the optimal condition for the FPI produced by the pH-shift method from the Tra catfish dark muscle.

#### 3.1.3. The Effects of Extraction Time (Trial 3)

PER, FPI-PR, and FPI-DMR increased when the extraction time was increased from 30 min to 60 min, followed by stable values from 60 min to 120 min, reaching a maximum at 150 min and decreasing after that (Figure 5). An increase in PER when the extraction time was extended from 20 min to 120 min was reported by Batista [55]. Increased FPI-PR and FPI-DMR occurred when the extraction time was extended from 30 to 75 min in alkali-aided protein extraction from catfish by-products [54].

The denaturation of proteins resulted in decreased PER, FPI-PR and FPI-DMR when the extraction time was prolonged to 180 min in this study, but extending the extraction time may promote the denaturation, as seen by the aggregation and polymerization of myofibrillar/cytoskeletal proteins in alkaline treatments [56]. Some proteins may thus unfold under alkaline conditions, forming hydrophobic interactions and disulfide bonds. These formations link protein chains together, resulting in protein aggregations and polymerization [57,58,59]. These protein fractions can be retained together in the sediment during acid precipitation [58].

### 3.2. Comparison of the FPI Produced from Different Rest Raw Materials and Surimi

#### 3.2.1. FPI Processing Effectivity

FPI-PR is a good indicator of the economic feasibility of the pH-shift method. In previous studies, FPI-PR has been observed to range between 42% and 90%, depending on methods used to measure protein concentration, fish species, types of raw material, centrifugation force, and the content of water-soluble sarcoplasmic proteins [60]. The FPI-PR and FPI-DMR were highest in the dark muscle (88.9 ± 5.3% and 34.3 ± 0.2%, respectively), followed by the ACO (83.0 ± 2.9% and 32.7 ± 2.7%), and lowest in the HBB (68.2 ± 4.8% and 19.1 ± 1.4%). Although these different side streams had similar crude protein contents (Table 2), the protein composition differed. The HBB may contain high residual blood containing water-soluble haemoglobin and myoglobin and stromal proteins [61] which are lost during water processes [60]. The heme proteins were removed mostly from the FPI production using the alkaline pH-shift method—as Abdollahi et al. (2016), Kristinsson, Theodore, Demir and Ingadottir [52] observed—resulting in a lower protein recovery. However, this is a desirable feature in FPI production because the residual heme proteins in FPI act as the main pro-oxidants, causing lipid and protein oxidation [46,62,63]. At the same time, stroma proteins are not soluble, regardless of the pH or ionic strength of the solution [52,64]. Therefore, these proteins remained in the bone and skin fractions (sediment). Additionally, some proteins may not have been recovered because they were still stuck to the bone fractions and lost in the sediment in the HBB-FPI processing [60], or could have been lost in the top lipid layer after centrifugation [52]. Zhang and Chang [54] also showed a low FPI-DMR (below 17%) during protein recovery from the catfish head and frame blend.

Most lipids were removed during FPI production (90.4%–95.2%), depending on the lipid content of the raw materials (Table 1), indicating the high capacity of the pH-shift processing in separating and removing lipids from raw materials. The lipids are separated during the pH-shift process based on their density and polarity [52]. At pH 12, proteins were solubilized and separated from the storage lipids and membrane phospholipids [65]. Most of the storage lipids may come to the surface in the first centrifugation due to their lower density, while most unsaturated membrane phospholipids may be separated into the first sediment [66].

Ash content was reduced by 86.3 to 91.5% during FPI production (Table 1). In the HBB, the ash was removed mainly from the insoluble bone fraction separated into the first sediment due to the bony property of the raw material. The highest mineral removal was indicated in the HBB-FPI production, reflecting that most of the bone fraction was removed from the FPI. Ash reduction in the dark muscle and ACO-FPI may be due to the release of blood containing iron and heme proteins in the water phase during FPI production [46]. Impurities with high mineral content could possibly be used in animal feeds as a mineral additive [67] or produced for collagen/gelatin, orthopedics, or dental materials (for example, the HBB fraction with a high level of Ca and p), as suggested by Nam, Van Hoa, Anh and Trung [19].

#### 3.2.2. Proximate Composition

The crude protein content of the FPIs was much higher than in the corresponding raw materials, although the water content was also higher (Table 2). The alkaline pH-shift process thus appeared to effectively separate insoluble impurities (bone, skin and connective tissues) and lipids. The trends of the changes in the proximate composition of the FPI products compared to their raw material agreed with studies performed on salmon and herring by-products [50,66] and cod by-products [50]. The lipid and ash contents of the FPIs were much lower compared to their corresponding raw materials. This is because most of the lipids and ash were removed during FPI processing, as discussed above. Lipid removal from the FPI is important because muscle lipids are susceptible to oxidation, leading to rancidity [46,56,60].

Significantly higher lipid content in the surimi than the FPIs may be due to the higher lipid content of the raw material (including trimmings with high lipid content) and lower lipid removal in the water washing process in surimi production. Membrane lipids are retained during the water washing process and the storage lipids co-aggregate with proteins [65]. A lower efficiency of lipid-lowering (82%) by water washing than the alkaline pH-shift method of FPI production (94%) has also been observed in broiler meat [65], Atlantic croakers [56], and catfish [52]. Lower protein content in the surimi may reflect lower protein recovery during surimi processing as compared to FPI processing [52]. Only myofibrillar proteins are recovered in surimi processing, while water-soluble proteins (sarcoplasmic proteins) are removed into the water streams [68]. Additionally, some muscle proteins are attached to bone and skin and are lost in the separation step [60].

Although the three raw side stream materials examined had different lipid contents, the corresponding FPIs all had a similar lipid content. This result may reflect that the remaining lipids in the FPIs were protein-linked lipids. Both membrane lipids and storage lipids are removed effectively after centrifugation during FPI processing, as Kristinsson et al. (2013) observed. The FPI products produced from the dark muscle and ACO had a similar proximate composition. These two fractions could be combined to produce FPI, unless their functional properties differ.

#### 3.2.3. Amino Acid Profiles

The amino acid profiles of the FPIs influence both nutritional value and functional properties. All three FPIs had a similar amino acid composition (Table 3). Glutamic acid, aspartic acid, lysine and leucine were the primary components of all FPIs, similar to FPIs obtained from gutted herring [69] and rainbow trout [67]. Essential amino acids accounted for almost 50% of the total amino acids. All FPIs basically complied well with the FAO/WHO/UNU [70] recommendations for adults, indicating that these FPIs may be useful as additives or ingredients in developing proteinaceous food products for adults needing essential amino acids [67].

Hydroxyproline was not detected in FPI obtained from the dark muscle and HBB, and only 0.6 g/100 g protein was present in ACO. This amino acid is primarily found in collagen. The collagen fraction was thus effectively removed during the pH-shift processing, in agreement with the study by Marmon and Undeland [69]. The cysteine/cystine content was also comparable with the herring FPI studied by [69], with content levels from 1.0 to 1.1 g/100 g protein.

The residual cysteine/cystine and hydrophobic amino acids play an important role in the FPI application. Under food processing treatments such as heating and mechanical shear, proteins’ tertiary structure may be unfolded, exposing sulfhydryl, cystine and hydrophobic groups and forming disulfide bonds and hydrophobic interactions. These chemical activities contribute to the gel-forming ability of FPIs. The FPIs might be used in gelling-based food products, similar to surimi or mince [57,71]. However, the natural disufide bond and hydrophobic interaction formations in FPIs during storage should be avoided, as this could lead to aggregation and decrease their gel-forming ability [72].

#### 3.2.4. SDS-PAGE

Tropomyosin, a protein with a molecular weight (MW) ranging from 35 to 40 kDa, was dominant in the FPIs produced. Other protein bands were obtained in ranges of 70–100 kDa, 140–260 kDa, 40–50 kDa, and approximately 15 kDa (Figure 6). A different protein pattern was observed in FPIs made from Atlantic croaker, which showed myosin to be most abundant [56]. G-actin (46–49 kDa) and tropomyosin were the two main proteins recovered in the surimi, similar to the surimi produced from Atlantic croaker [56]. A higher intensity of myosin heavy chain and heavy meromyosin (140–260 kDa) were observed in the commercial surimi than in the FPIs, while protein bands from 70 to 100 kDa were only identified in the FPIs. Several studies have shown the protein pattern of surimi to be similar to that of its raw material [59,73]. These results reflect that myosin may be partly hydrolysed during FPI processing to form lower molecular weight bands [46,47,56]. In addition, myosin heavy chain denaturation and aggregation might already occur in FPIs, leading to the loss of protein extractability and missing myosin heavy chain bands in FPIs [59,72]. Myosin is well known as being primarily responsible for the functionality of muscle foods, especially gelation [71]. The degradation of myosin heavy chain and heavy meromyosin in the FPIs may lower their gel-forming ability. Small cysteine-containing proteins or microbial transglutaminase additions should be considered in cases using these FPIs in gelling-based products [71,74]. The intensity of the protein bands varied slightly across the samples. A protein band ranging from 15 to 25 kDa (myosin light chain) was clearly identified for ACO-FPI, but was weak in DM-FPI and HBB-FPI.

#### 3.2.5. Colour

When compared to the surimi, all FPIs produced had lower lightness and whiteness values and higher yellowness and redness components, except for the redness of the ACO-FPI (Table 4). A similar result was observed in surimi and protein isolates made from mackerel (*Rastrelliger brachysoma*) [75]. Colour can be influenced by the amount of residual blood and dark muscle, as well as the presence of pigments such as heme proteins and melanin [76,77]. Conventional washing during surimi processing may reduce the pro-oxidants such as heme proteins and metal ions which cause colour changes [12,46,68,78].

Lower whiteness and higher yellowness and redness were likely due to blood residue and the oxidation of heme proteins (especially myoglobin) in the recovered proteins catalysed by the alkaline pH [46,75]. The HBB raw material may have contained high residual blood content due to the presence of the main blood vasculature along the backbone. Previous studies have shown that the alkaline and acid conditions during FPI processing promote the auto-oxidation of heme proteins and the discolouration of FPIs [12,75]. The presence of blood and/or residual heme proteins in fish side streams and FPIs produced from them can thus affect several quality properties, such as oxidative stability and whiteness [46,62]. The FPI produced from ACO was the lightest and of the highest whiteness, while HBB-FPI had the lowest. The ACO-FPI was produced from ACO, which may have had a low content of heme proteins, thereby resulting in the lowest redness value [79]. The high redness value in the DM-FPI may have been due to high levels of residual heme proteins because the dark muscle material had a high heme protein content, as has also been shown in mackerel and trout [79]. Among the FPIs the ACO-FPI is most promising as a food ingredient because of its high whiteness. However, prewashing the raw material before solubilization and a high water ratio used during homogenizing with the raw material may improve the whiteness of the FPIs even further, as suggested by Abdollahi, Marmon, Chaijan and Undeland [46].

## 4. Conclusions

The objective of the study was to investigate the potential for using different Tra catfish side streams for fish protein isolate production by assessing the chemical composition and characteristics of the side streams and the optimization of the protein isolate extraction conditions.

The pH of the extraction solution had a substantial effect on FPI production. In contrast, the ratio of extraction solution to raw material had a relatively minor effect on the FPI process. The protein extracted at pH 12 with a ratio of 8 (extraction solution/raw material, volume/weight) for 150 min gave the best results.

Most of the lipid and ash content was removed in the production of the FPIs, resulting in high protein content. The FPIs had a higher protein content and a lower lipid and ash content than the industrial surimi. All the FPIs had good amino acid compositions that could be used in food products for adults. In addition, with high protein and dry matter recoveries, Tra catfish dark muscle and ACO are potential sources for producing protein isolates using the pH-shift method. However, the prewashing and homogenisation steps should be studied to improve the whiteness of the FPIs.

The FPIs produced from dark muscle and ACO had similar chemical properties but different colour attributes. These raw materials should thus be processed into FPIs separately, adjusting each FPI towards the production of a specific value-added food product. However, other physicochemical properties, such as texture attributes, gel-forming ability, and lipid stability, should be studied further.

Overall, the study showed that Tra catfish FPIs have great potential as food ingredients in gel-based snack foods, similar to surimi and mince, and can be used as ingredients in the development of other higher-value products.

## Figures and Tables

**Figure 1 foods-11-01531-f001:**
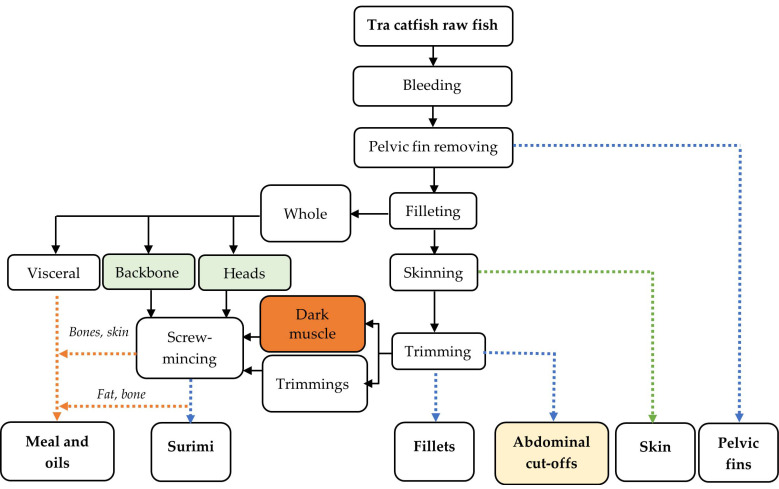
Industrial Tra catfish processing streams. The blue line indicates the stream producing products for human consumption, the orange represents the stream used to produce animal feed, and the green colour shows material for producing functional products. Colour-filled boxes indicate the material collected for optimization protein recovery by the pH-shift method.

**Figure 2 foods-11-01531-f002:**
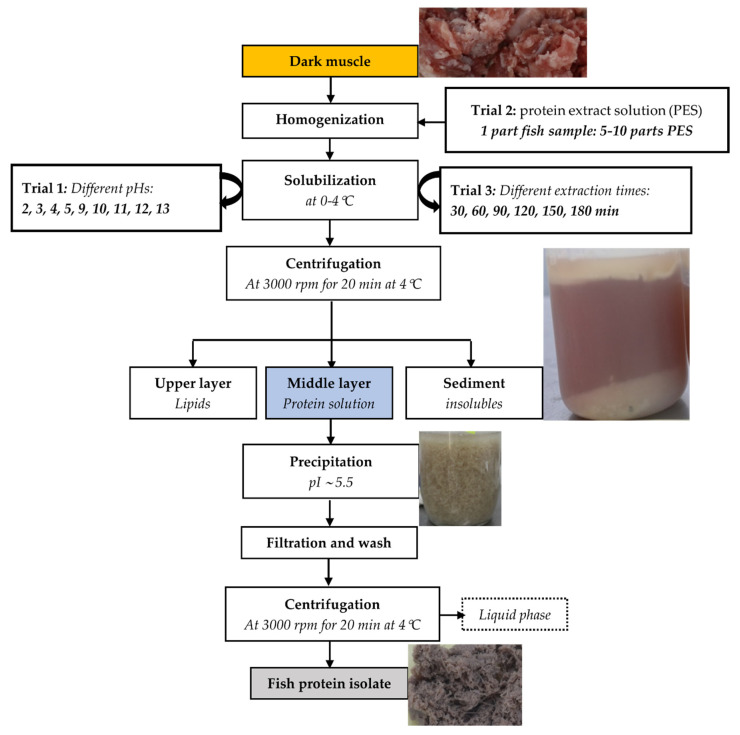
Experimental design for the optimization of protein recovery from Tra catfish dark muscle, carried out through 3 trials. Orange-filled box: protein and water content measured; blue-filled box: weight of the protein solution recorded and its protein content determined; grey-filled box: weight recorded, water and protein content determined.

**Figure 3 foods-11-01531-f003:**
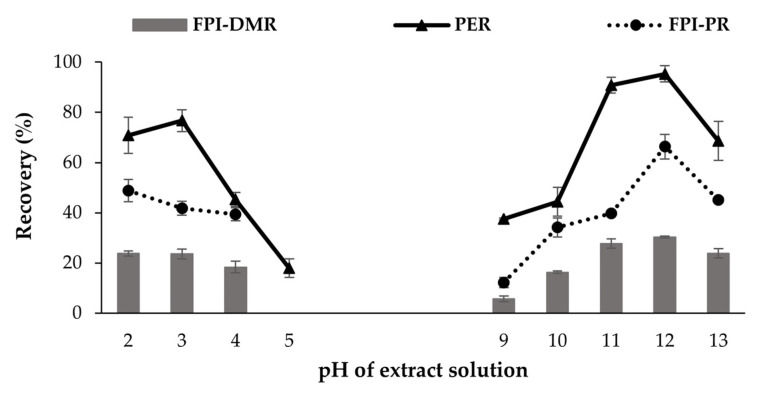
Effect of pH on protein extraction recovery (PER, %), FPI protein recovery (FPI-PR, %), and dry matter recovery (FPI-DMR, %) of the fish protein isolate produced from Tra catfish dark muscle. The protein extraction was carried out for 60 min with a volume/weight extraction ratio of 8.

**Figure 4 foods-11-01531-f004:**
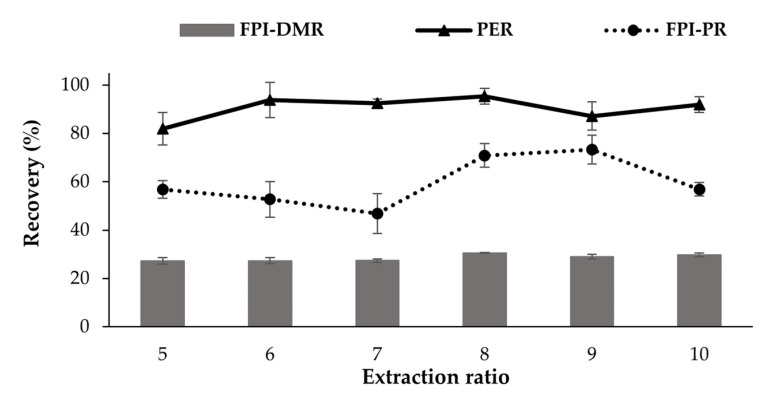
Effect of extraction ratio (extract solution volume/raw material weight) on PER (%), FPI-PR (%), and FPI-DMR (%) of the fish protein isolate (FPI) produced from the Tra catfish dark muscle. The protein extraction was carried out at pH 12 for 60 min.

**Figure 5 foods-11-01531-f005:**
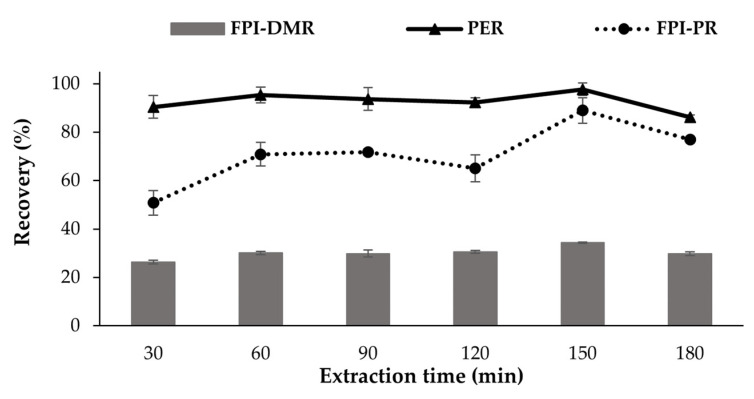
Effect of extraction time on the PER (%), FPI-PR (%), and FPI-DMR (%) of the fish protein isolate (FPI) produced from Tra catfish dark muscle. The proteins were extracted at pH 12 with an extraction ratio of 8.

**Figure 6 foods-11-01531-f006:**
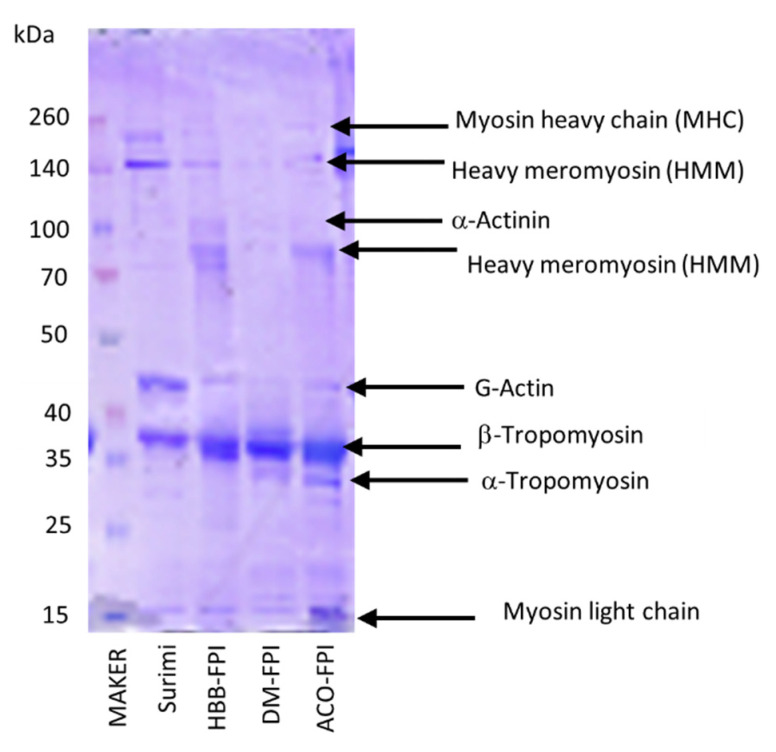
Sodium dodecyl sulfate-polyacrylamide gel electrophoresis (SDS-PAGE) patterns of surimi and FPI made from head and backbone blend (HBB-FPI), dark muscle (DM-FPI), and abdominal cut-offs (ACO-FPI).

**Table 1 foods-11-01531-t001:** Process effectivity of the fish protein isolate (FPI) processing from Tra catfish dark muscle (DM-FPI), head and backbone blend (HBB-FPI), abdominal cut-offs (ACO-FPI), and surimi made from the same batch of raw material. Results are expressed as means ± SD of triplicate measurements (*n* = 3) *.

Production	FPI-PR (%)	FPI-DMR (%)	Lipid Removal (%)	Ash Removal (%)
DM-FPI	88.9 ± 5.3 ^a^	34.3 ± 0.2 ^a^	90.4 ± 1.0 ^b^	86.3 ± 0.8 ^b^
HBB-FPI	68.2 ± 4.8 ^b^	19.1 ± 1.4 ^b^	95.2 ± 0.7 ^a^	91.5 ± 0.6 ^a^
ACO-FPI	83.0 ± 2.9 ^a^	32.7 ± 2.7 ^a^	93.6 ± 0.2 ^a^	87.2 ± 0.5 ^b^

* Different superscript letters show significant differences within column at *p* < 0.05.

**Table 2 foods-11-01531-t002:** Proximate composition (%) of the raw materials and fish protein isolates (FPIs) produced from Tra catfish dark muscle, head and backbone blend (HBB), abdominal cut-offs (ACO), and surimi made from the same batch of raw materials. Results are expressed as means ± SD of triplicate measurements (*n* = 3) ^*^.

		Water Content	Crude Protein	Lipid Content	Ash Content
** *Raw material* **	Dark muscle	66.5 ± 1.0 ^An^	14.7 ± 0.2 ^Bn^	17.6 ± 1.5 ^Bm^	0.8 ± 0.1 ^Bm^
	HBB	54.8 ± 1.0 ^Cn^	15.2 ± 0.1 ^An^	21.9 ± 0.6 ^Am^	7.8 ± 0.4 ^Am^
	ACO	60.8 ± 2.0 ^Bn^	15.3 ± 0.1 ^An^	23.5 ± 1.1 ^Am^	1.0 ± 0.1 ^Bm^
**FPIs and surimi**	DM-FPI	73.9 ± 0.7 ^bcm^	23.5 ± 0.9 ^am^	3.2 ± 0.0 ^bn^	0.1 ± 0.0 ^bn^
	HBB-FPI	77.3 ± 0.8 ^am^	20.4 ± 0.5 ^bm^	2.8 ± 0.4 ^bn^	0.1 ± 0.0 ^bn^
	ACO-FPI	73.1 ± 2.1 ^cm^	24.4 ± 1.4 ^am^	3.1 ± 0.1 ^bn^	0.0 ± 0.0 ^bn^
	Surimi	77.0 ± 0.1 ^ab^	17.5 ± 0.3 ^c^	5.4 ± 0.1 ^a^	0.2 ± 0.0 ^a^

* Different uppercase superscript letters indicate significant differences within the column for the raw material; different lowercase superscript letters ^a–c^ show significant differences within the column for the FPIs and surimi; different lowercase superscript letters ^m–n^ indicate significant differences between raw material and the corresponding FPI at same parameter at *p* < 0.05.

**Table 3 foods-11-01531-t003:** Amino acid composition (g/100 g crude protein) of the fish protein isolates produced from Tra catfish dark muscle (DM-FPI), head and backbone blend (HBB-FPI), and abdominal cut-offs (ACO-FPI).

Amino Acids	DM-FPI	HBB-FPI	ACO-FPI	FAO/WHO/UNU [70] *
Hydroxyproline	ND	ND	0.6	
Alanine ^b^	6.3	6.5	6.4	
Arginine ^a^	6.6	6.9	6.3	
Aspartic acid	**11.2**	**12.1**	**10.5**	
Cysteine/Cystine	1.1	1.1	1.0	
γ-Aminobutyric acid	ND	ND	ND	
Glutamic acid	**18.4**	**19.4**	**17.2**	
Glycine ^b^	3.7	3.9	5.1	
Histidine ^a^	2.6	2.6	2.8	1.5
Isoleucine ^ab^	4.7	5.0	4.6	3.0
Leucine ^ab^	**8.8**	**9.1**	**7.9**	5.9
Lysine ^a^	**9.3**	**10.1**	**8.8**	4.5
Methionine ^ab^	3.3	3.5	2.9	2.2
Phenylalanine ^ab^	3.8	4.2	3.8	3.8
Proline ^b^	4.6	4.3	5.0	
Serine	4.2	4.4	4.1	
Threonine ^a^	4.6	4.9	4.2	2.3
Tyrosine	2.2	3.5	3.1	
Valine ^ab^	5.0	5.3	4.8	3.9
Total amino acids	100.5	106.8	99.1	
Total essential amino acids	48.8	51.7	46.2	
Total Hydrophobic amino acid	40.3	41.8	40.6	

^a^ Essential amino acid for infants. ^b^ Hydrophobic amino acids. * FAO/WHO/UNU recommendations for adults (2007).

**Table 4 foods-11-01531-t004:** Colour of the fish protein isolates produced from Pangasius dark muscle, head and backbone blend (HBB), abdominal cut-offs (ACO), and surimi. Results are expressed as means ± SD of triplicate measurements (*n* = 3) *.

		Lightness (*L**)	Redness (*a**)	Yellowness (*b**)	Whiteness
** *FPIs* **	DM-FPI	47.8 ± 1.0 ^cA^	1.8 ± 0.1 ^bA^	11.4 ± 0.5 ^aA^	46.6 ± 1.1 ^cB^
	HBB-FPI	34.0 ± 4.1 ^dC^	**3.9 ± 1.6 ^aC^**	5.4 ± 2.0 ^cB^	33.6 ± 4.0 ^dC^
	ACO-FPI	**64.9 ± 0.3 ^bB^**	−1.2 ± 0.1 ^cB^	**11.7 ± 0.2 ^aA^**	**63.0 ± 0.4 ^bA^**
** *Industrial surimi* **		**73.5 ± 0.6 ^a^**	−0.2 ± 0.1 ^c^	4.2 ± 0.5 ^c^	**73.2 ± 0.6 ^a^**

* Different uppercase superscript letters indicate significant differences within the column; different lowercase superscript letters show significant differences within the column between FPI products at *p* < 0.05.

## Data Availability

The data presented in this study are available on request from the corresponding author.

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
