# Peer review of "Protein Recovery of Tra Catfish (Pangasius hypophthalmus) Protein-Rich Side Streams by the pH-Shift Method"

_foods, 2022, doi:10.3390/foods11111531_

Round 1

Reviewer 1 Report

  1. Gel-forming ability of protein isolate is missing. In case you want to compare with surimi, the gel-forming ability (e.g. breaking force, deformation, whiteness, and expressible drip) is needed. Since the authors concluded that Tra catfish FPIs can be used as food ingredients in gel-based snack foods, similar to surimi and mince, or used as ingredients in the development of other higher-value products.
  2. Line 160, why the precipitation pH was at 5.5? How was the solubility profile of these raw materials?
  3. Line 246 and 465. It should read as “SDS-PAGE”
  4. Line 266-273 and throughout the text. Please italicize L* a* and b*.
  5. Line 357-361. Authors stated that “The denaturation of proteins may result in decreased PER, FPI-PR and FPI-DMR when the extraction time was prolonged to 180 min in this study….” How about the aggregation and polymerization of proteins? as you can see in the SDS-PAGE.
  6. How about the heme-protein removal efficacy of the process? Since some raw materials are rich in dark muscle, blood and myoglobin.
  7. How about the oxidative stability of residual lipid and heme proteins of protein isolate?
  8. From Fig.6, myosin heavy chain band intensities of FPIs were almost absent when compared to surimi. This can definitely affect the functionality of the FPIs. Please discuss on this point.
  9. Table 4. It is wrong for “Yellowness (a*) and Redness (b*)”. a* is for redness and b* is for yellowness. Please recheck.
  10. Conclusion is too long. Line 516-525 is not conclusion. It can be found in Introduction already. So, delete it all and please make it concise and informative.

Reviewer 2 Report

Dear Authors,

Although this subject don´t present novelty it has good scientifical quality since the recovery of proteins and their directly as food or processes into fish-based protein food is of great interest. The use of principles of circular economy in the seafood sector is essential.

Moreover, the English language is of excellent quality and the main ideas are clearly understandable. Presentation quality is also fine. 

Round 2

Reviewer 1 Report

All points raised by reviewers were carefully addressed and answered point-by-point. So, it can be accepted.